# Solving the Rubik's Cube with Approximate Policy Iteration

**Stephen McAleer**[*]
Department of Statistics
University of California, Irvine
`smcaleer@uci.edu`

**Forest Agostinelli**[*]
Department of Computer Science
University of California, Irvine
`fagostin@uci.edu`

**Alexander Shmakov**[*]
Department of Computer Science
University of California, Irvine
`ashmakov@uci.edu`

**Pierre Baldi**
Department of Computer Science
University of California, Irvine
`pfbaldi@ics.uci.edu`

## Abstract

Recently, Approximate Policy Iteration (API) algorithms have achieved super-human proficiency in two-player zero-sum games such as Go, Chess, and Shogi without human data. These API algorithms iterate between two policies: a slow policy (tree search), and a fast policy (a neural network). In these two-player games, a reward is always received at the end of the game. However, the Rubik's Cube has only a single solved state, and episodes are not guaranteed to terminate. This poses a major problem for these API algorithms since they rely on the reward received at the end of the game. We introduce Autodidactic Iteration: an API algorithm that overcomes the problem of sparse rewards by training on a distribution of states that allows the reward to propagate from the goal state to states farther away. Autodidactic Iteration is able to learn how to solve the Rubik's Cube without relying on human data. Our algorithm is able to solve 100% of randomly scrambled cubes while achieving a median solve length of 30 moves — less than or equal to solvers that employ human domain knowledge.

## 1 Introduction

The Rubik's Cube is a classic combination game that poses unique and interesting challenges for AI and machine learning. Although the state space is astronomically large ($4.2 \times 10^{19}$ different states for the 3x3x3 cube), only a single state is considered solved. Furthermore, unlike the game of Go or Chess, the Rubik's Cube is a single-player game and a sequence of random moves, no matter how long, is unlikely to end in the solved state. Developing reinforcement learning algorithms to deal with this property of the Rubik's Cube might provide insight into other sparse-reward environments. While methods for solving the Rubik's Cube have been developed, only relatively recently have methods been derived that can compute the minimal number of moves required to solve the cube from any given starting configuration (Rokicki et al., 2014; Rokicki, 2014). In addition, the Rubik's Cube and its solutions are deeply rooted in group theory, raising interesting and broader questions about the applicability of machine learning methods to complex symbolic systems, including mathematics. Finally, the classical 3x3x3 Rubik's Cube is only one representative of a much larger family of possible combination puzzles, broadly sharing the characteristics described above and including: (1) Rubik's Cubes with longer edges (e.g. 4x4x4); (2) Rubik's Cubes in higher dimensions (e.g. 2x2x2x2); as well as (3) Rubik's cube on non-cubic geometries (Pyriminix, etc) and their combinations. As the length of the sides and dimensions are increased, the complexity of the underlying combinatorial problems rapidly increases and, for instance, God's numbers for the 4x4x4 cube is not known. In short, for all these reasons, the Rubik's Cube and its variations pose interesting challenges for machine learning. Here we develop deep reinforcement learning methods, in particular a new form of Approximate Policy Iteration (API), for addressing these challenges.

---

[*]Equal contribution

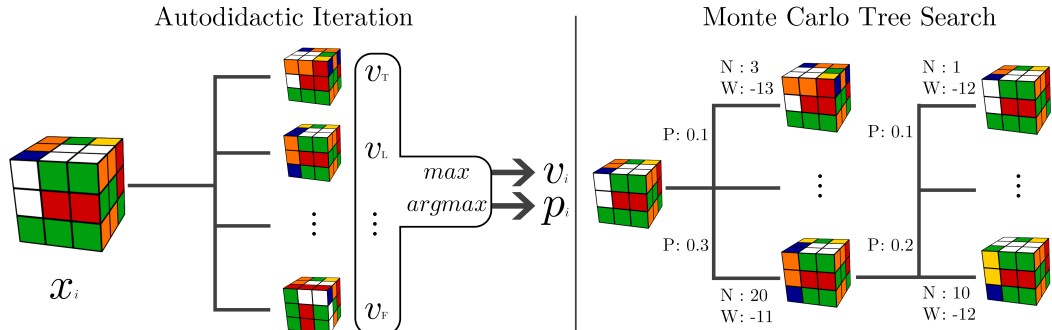

Figure 1: An illustration of DeepCube. The training and solving process is split up into ADI and MCTS. First, we iteratively train a DNN by estimating the true value of the input states using breadth-first search. Then, using the DNN to guide exploration, we solve cubes using Monte Carlo Tree Search. See methods section for more details.

Approximate Policy Iteration (Bertsekas & Tsitsiklis, 1995; Bagnell et al., 2004; Kakade & Langford, 2002; Scherrer, 2014; Lazaric et al., 2010) is a core Reinforcement Learning (RL) algorithm. Recently, a new type of API algorithm, called Dual Policy Iteration (Sun et al., 2018), has achieved success in two player zero-sum games such as Go, Chess, Shogi and Hex (Anthony et al., 2017; Silver et al., 2017a;c). AlphaZero (Silver et al., 2017c) and ExIt (Anthony et al., 2017) are examples of Dual Policy Iteration. These algorithms update the policy in a more sophisticated way than traditional API methods by using two policies: a fast policy (usually a neural network) and a slow policy (usually tree search). The fast policy is trained via supervised learning on data generated from gameplay from the slow policy. The slow policy then uses the fast policy to guide the tree search. In this way, the fast policy is used to improve the slow policy, and the slow policy generates better data to train the fast policy.

This work is the first to solve the Rubik's Cube with reinforcement learning. Although DPI works for two-player games, our work is the first DPI algorithm to succeed in an environment with a high number of states and a small number of reward states. A number of these "needle in a haystack" environments such as generating meaningful sentences or code are seen as long-term goals for the field of AI. There is no clear way to apply current DPI algorithms such as AlphaZero or ExIt to sparse-reward environments such as the Rubik's Cube. This is because a randomly initialized policy will be unlikely to encounter the single reward state. This will cause the value function to be biased or divergent and the fast policy will not converge to the optimal policy. In our work we overcome this problem by training the fast policy on a distribution of states that propagate the reward signal to from the goal state to the further states.

Our algorithm, called Autodidactic Iteration (ADI), trains a neural network value and policy function through an iterative process. These neural networks are the "fast policy" of DPI described earlier. In each iteration, the inputs to the neural network are created by starting from the goal state and randomly taking actions. The targets seek to estimate the optimal value function by performing a breadth-first search from each input state and using the current network to estimate the value of each of the leaves in the tree. Updated value estimates for the root nodes are obtained by recursively backing up the values for each node using a max operator. The policy network is similarly trained by constructing targets from the move that maximizes the value. After the network is trained, it is combined with MCTS to effectively solve the Rubik's Cube. We call the resulting solver DeepCube.

## 2 RELATED WORK

Erno Rubik created the Rubik's Cube in 1974. After a month of effort, he came up with the first algorithm to solve the cube. Since then, the Rubik's Cube has gained worldwide popularity and many human-oriented algorithms for solving it have been discovered  (Ruwix). These algorithms are simple to memorize and teach humans how to solve the cube in a structured, step-by-step manner.

Human-oriented algorithms to solve the Rubik's Cube, while easy to memorize, find long suboptimal solutions. Since 1981, there has been theoretical work on finding the upper bound for the number

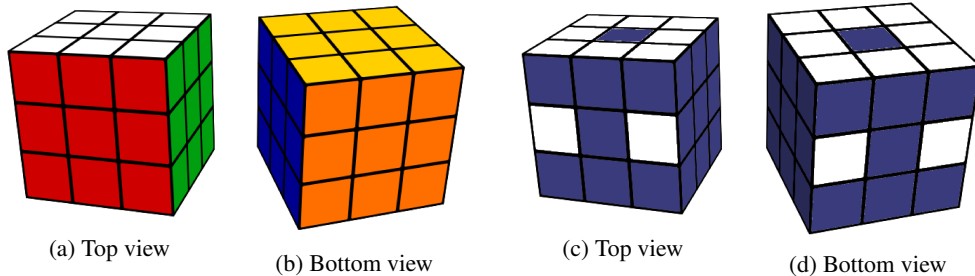

(a) Top view    (b) Bottom view    (c) Top view    (d) Bottom view

Figure 2: Visualizations of the Rubik's Cube. (a) and (b) show the solved cube as it appears in the environment. (c) and (d) show the cube reduced in dimensionality for input into the DNN. Stickers that are used by the DNN are white, whereas ignored stickers are dark.

of moves necessary to solve any valid cube configuration  (Thistlethwaite, 1981; Reid, 1995; Radu, 2007; Kunkle & Cooperman, 2007; Rokicki, 2010). Finally, in 2014, it was shown that any valid cube can be optimally solved with at most 26 moves in the quarter-turn metric, or 20 moves in the half-turn metric  (Rokicki et al., 2014; Rokicki, 2014). The quarter-turn metric treats 180 degree rotations as two moves, whereas the half-turn metric treats 180 degree rotations as one move. For the remainder of this paper we will be using the quarter-turn metric. This upper bound on the number of moves required to solve the Rubik's cube is colloquially known as God's Number.

Algorithms used by machines to solve the Rubik's Cube rely on hand-engineered features and group theory to systematically find solutions. One popular solver for the Rubik's Cube is the Kociemba two-stage solver  (Kociemba). This algorithm uses the Rubik's Cube's group properties to first maneuver the cube to a smaller sub-group, after which finding the solution is trivial. Heuristic based search algorithms have also been employed to find optimal solutions. In 1985, it was shown that iterative deepening A* (IDA*) could be used to find optimal solutions to combination puzzles (Korf, 1985). This algorithm was then combined with pattern databases (Culberson & Schaeffer, 1998) to successfully find optimal solutions to the Rubik's Cube (Korf, 1997). Later, (Sturtevant & Chen, 2016) used advances in compute power (5TB of external memory and parallel processing), to show that bidirectional brute-force search could find optimal solutions to the Rubik's Cube in up to 28 hours. More recently, there has been an attempt to train a DNN – using supervised learning with hand-engineered features – to act as an alternative heuristic (Brunetto & Trunda, 2017). These search algorithms, however, take an extraordinarily long time to run and usually fail to find a solution to randomly scrambled cubes within reasonable time constraints. Besides hand-crafted algorithms, attempts have been made to solve the Rubik's Cube through evolutionary algorithms  (Smith et al., 2016; Lichodzijewski & Heywood, 2011). However, these learned solvers can only reliably solve cubes that are up to 5 moves away from the solution.

We solve the Rubik's Cube using API (Bertsekas & Tsitsiklis, 1995; Bagnell et al., 2004; Kakade & Langford, 2002; Scherrer, 2014; Lazaric et al., 2010). API is a general purpose reinforcement learning algorithm, and includes API with stochastic policies  (Schulman et al., 2015; Kakade, 2001; Bagnell & Schneider, 2003; Baxter & Bartlett, 2001), conservative API  (Kakade & Langford, 2002), and API with learned critics  (Rummery & Niranjan, 1994). Even though many of these API algorithms have certain theoretical optimality guarantees, they rely on random exploration (such as REINFORCE (Williams, 1992)-type policy gradient or $\epsilon$-greedy) and as a result are not sample-efficient.

Dual Policy Iteration (Sun et al., 2018) algorithms such as AlphaZero and Exit are a new type of API algorithm that has achieved success in two player zero-sum games such as Go, Chess, Shogi and Hex (Anthony et al., 2017; Silver et al., 2017a;c) . These algorithms use two policies: a fast policy and a slow policy. The fast policy is trained via supervised learning on data generated from gameplay from the slow policy. The slow policy then uses the fast policy to guide the tree search. The main difference between DPI and traditional API algorithms is the use of a forward dynamics model. With the use of this model, the slow policy can perform a tree search, abandoning the need for random exploration. Using this forward dynamics model allows for improved sample efficiency compared to traditional API algorithms.

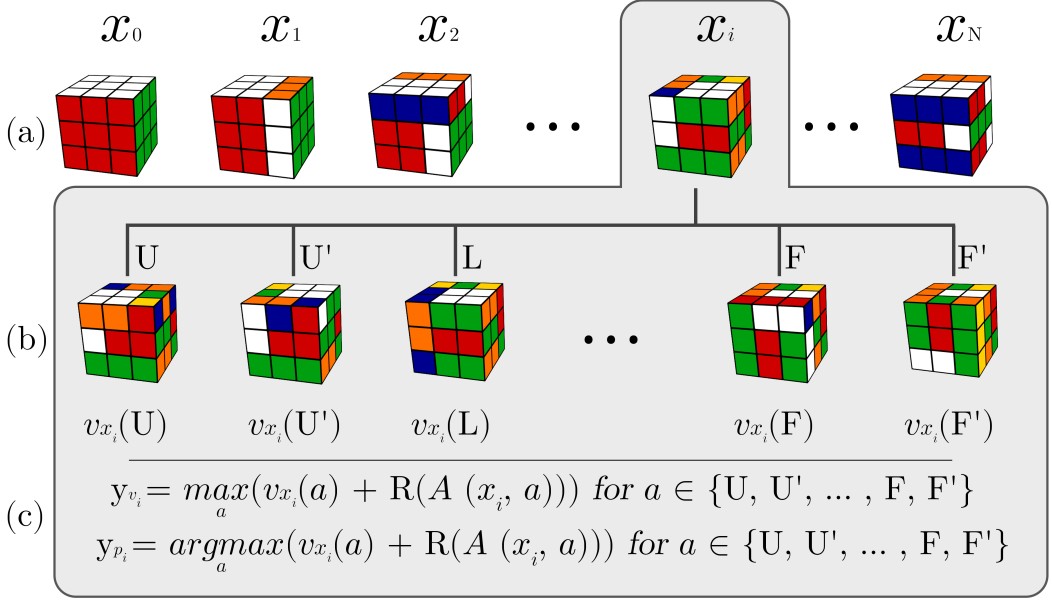

Figure 3: Visualization of training set generation in ADI. (a) Generate a sequence of training inputs starting from the solved state. (b) For each training input, generate its children and evaluate the value network on all of them. (c) Set the value and policy targets based on the Bellman Operator.

It is now well-known that reinforcement learning with nonlinear function approximation is prone to diverge (Baird, 1995; Boyan & Moore, 1995; Gordon, 1995). Wang & Dietterich (1999) approaches the problem of learning a stable value function with nonlinear approximation for combinatorial optimization by using a composite error function. Learning a value function with nonlinear approximation within API is not much of a problem when trying to solve Go or Chess. This is because the problem is simply binary classification based on the current board state. With the Rubik's Cube, on the other hand, approximating the value function is very difficult because even a naive monte-carlo approach would not work since it will never encounter the solved state. ADI ensures stable and approximately unbiased training of the value function by sampling states in a way that ensures the goal state's learning signal will propagate during the training. This approach is somewhat similar to the method proposed in Boyan & Moore (1995).

## 3 THE RUBIK'S CUBE

The Rubik's Cube consists of 26 smaller cubes called cubelets. These are classified by their sticker count: center, edge, and corner cubelets have 1, 2, and 3 stickers attached respectively. There are 54 stickers in total with each sticker uniquely identifiable based on the type of cubelet the sticker is on and the other sticker(s) on the cubelet. Therefore, we may use a one-hot encoding for each of the 54 stickers to represent their location on the cube. However, because the position of one sticker on a cubelet determines the position of the remaining stickers on that cubelet, we may actually reduce the dimensionality of our representation by focusing on the position of only one sticker per cubelet. We ignore the redundant center cubelets and only store the 24 possible locations for the edge and corner cubelets. This results in a 20x24 state representation which is demonstrated in Figures 2c and 2d.

Moves are represented using face notation originally developed by David Singmaster: a move is a letter stating which face to rotate. *F*, *B*, *L*, *R*, *U*, and *D* correspond to turning the *front*, *back*, *left*, *right*, *up*, and *down* faces, respectively. A clockwise rotation is represented with a single letter, whereas a letter followed by an apostrophe represents a counter-clockwise rotation. For example: *R* and *R'* would mean to rotate the *right* face 90° clockwise and counter-clockwise, respectively. Formally, the Rubik's Cube environment consists of a set of $4.3 \times 10^{19}$ states $\mathcal{S}$ which contains one special state, $s_{solved}$, representing the goal state. At each timestep, $t$, the agent observes a state $s_t \in \mathcal{S}$ and takes an action $a_t \in \mathcal{A}$ with $\mathcal{A} := \{F, F', \ldots, D, D'\}$. After selecting an action, the agent observes a new

state $s_{t+1} = A(s_t, a_t)$ and receives a scalar reward, $R(s_{t+1})$, which is 1 if $s_{t+1}$ is the goal state and $-1$ otherwise. It is important to note that this environment is deterministic with known forward and backwards dynamics.

## 4 METHODS

We develop a novel algorithm called Autodidactic Iteration which is used to train a joint value and policy network. Once the network is trained, it is combined with MCTS to solve randomly scrambled cubes. The resulting solver is called DeepCube. Autodidactic Iteration is a DPI algorithm like AlphaZero and ExIt. However, since AlphaZero and ExIt cannot solve the Rubik's Cube, Autodidactic Iteration uses a state sampling procedure during training to prevent divergence of the value function and to ensure that the learning signal from the single reward state is propagated from the solved cube during training. In particular, the sampling distribution is generated by starting from the solved state and randomly taking actions, weighting cubes closer to the solved state more heavily. This sampling procedure is described in the State Sampling section below.

### 4.1 AUTODIDACTIC ITERATION

ADI is an iterative supervised learning procedure which trains a deep neural network $f_\theta(s)$ with parameters $\theta$ which takes an input state $s$ and outputs a value and policy pair $(v, \boldsymbol{p})$. The policy output $\boldsymbol{p}$ is a vector containing the move probabilities for each of the 12 possible moves from that state. Once the network is trained, the policy is used to reduce breadth and the value is used to reduce depth in the MCTS. In each iteration of ADI, training samples for $f_\theta$ are generated by starting from the solved cube. This ensures that some training inputs will be close enough to have a positive reward when performing a shallow search. Targets are then created by performing a depth-1 breadth-first search (BFS) from each training sample. The current value network is used to estimate each child's value. The value target for each sample is the maximum value and reward of each of its children, and the policy target is the action which led to this maximal value. Figure 3 displays a visual overview of ADI.

Formally, we generate training samples by starting with $s_{solved}$ and scrambling the cube $k$ times to generate a sequence of $k$ cubes. We do this $l$ times to generate $N = k * l$ training samples $X = [x_i]_{i=1}^N$. Each sample in the series has the number of scrambles it took to generate it, $D(x_i)$, associated with it. Then, for each sample $x_i \in X$, we generate a training target $Y_i = (y_{v_i}, \boldsymbol{y}_{p_i})$. To do this, we perform a depth-1 BFS to get the set of all children states of $x_i$. We then evaluate the current neural network at each of the children states to receive estimates of each child's optimal value and policy: $\forall a \in \mathcal{A}, (v_{x_i}(a), \boldsymbol{p}_{x_i}(a)) = f_\theta(A(x_i, a))$. We set the value target to be the maximal value from each of the children $y_{v_i} \leftarrow \max_a(R(A(x_i, a)) + v_{x_i}(a))$, and we set the policy target to be the move that results in the maximal estimated value $\boldsymbol{y}_{p_i} \leftarrow \text{argmax}_a(R(A(x_i, a)) + v_{x_i}(a))$. We then train $f_\theta$ on these training samples and targets $[x_i, y_{v_i}]_{i=1}^N$ and $[x_i, \boldsymbol{y}_{p_i}]_{i=1}^N$ to receive new neural network parameters $\theta'$. For training, we used the *RMSProp* optimizer (Hinton et al., 2014) with a mean squared error loss for the value and softmax cross entropy loss for the policy. Although we use a depth-1 BFS for training, this process may be trivially generalized to perform deeper searches at each $x_i$.

### 4.2 STATE SAMPLING

Correct sampling of training data is vital or the convergence of API. It is particularly important for the Rubik's Cube, since a naive state sampling will never include the solved state and the reward signal will not be learned by the value function. ADI trains on states that are generated by starting from the solved cube and taking random actions. This allows the reward signal to propagate from the solved state to states that are farther away. However, ensuring the reward signal is propagated is not the only consideration for the Rubik's Cube. Training a nonlinear approximate value function often leads to unstable or divergent behavior (Boyan & Moore, 1995; Baird, 1995; Gordon, 1995). This is because although the optimal value function is usually smooth and easy to approximate, the intermediate value functions generated by approximate policy iteration are rarely smooth, resulting in divergent behaviour. This problem is especially true for the Rubik's Cube since only one state has a positive reward. One solution to value function divergence is to only train on true value data. Such

---

**Algorithm 1:** Autodidactic Iteration

---

**Initialization:** $\theta$ initialized using Glorot initialization

**repeat**

    $X \leftarrow$ N scrambled cubes

    **for** $x_i \in X$ **do**

        **for** $a \in \mathcal{A}$ **do**

            $(v_{x_i}(a), \boldsymbol{p}_{x_i}(a)) \leftarrow f_\theta(A(x_i, a))$

        $y_{v_i} \leftarrow \max_a(R(A(x_i, a)) + v_{x_i}(a))$

        $\boldsymbol{y}_{p_i} \leftarrow \operatorname{argmax}_a(R(A(x_i, a)) + v_{x_i}(a))$

        $Y_i \leftarrow (y_{v_i}, \boldsymbol{y}_{p_i})$

    $\theta' \leftarrow train(f_\theta, X, Y)$

    $\theta \leftarrow \theta'$

**until** $iterations = M$;

---

a solution is presented by Boyan & Moore (1995), who only train the approximate value function on a subset of support states, whose value has been explicitly computed via tree search or similar methods. We take a similar approach by weighting states that are earlier in the rollout from the solved state more heavily during training. This way, the algorithm trains on data that it is more confident of the true value. We assign the loss weight $W(x_i) = \frac{1}{D(x_i)}$ to each sample $x_i$. Although since we are dealing with nonlinear function approximation and TD-learning we do not have any theoretical guarantees, empirically this state sampling procedure prevented divergent behaviour. Since the algorithm prioritized learning states close to the solved state, it was then able to use these predictions to come up with more accurate targets for states farther away. Without this novel state sampling procedure, Algorithm 1 would often diverge after several iterations, but with the state sampling procedure it did not diverge in any of our tests.

### 4.3 SOLVER

We employ an asynchronous Monte Carlo Tree Search augmented with our trained neural network $f_\theta$ to solve the cube from a given starting state $s_0$. We build a search tree iteratively by beginning with a tree consisting only of our starting state, $T = \{s_0\}$. We then perform simulated traversals until reaching a leaf node of $T$. Each state, $s \in T$, has a memory attached to it storing: $N_s(a)$, the number of times an action $a$ has been taken from state $s$, $W_s(a)$, the maximal value of action $a$ from state $s$, $L_s(a)$, the current virtual loss for action $a$ from state $s$, and $P_s(a)$, the prior probability of action $a$ from state $s$.

Every simulation starts from the root node and iteratively selects actions by following a tree policy until an unexpanded leaf node, $s_\tau$, is reached. The tree policy proceeds as follows: for each timestep $t$, an action is selected by choosing, $A_t = \operatorname{argmax}_a U_{s_t}(a) + Q_{s_t}(a)$ where $U_{s_t}(a) = cP_{s_t}(a)\sqrt{\sum_{a'} N_{s_t}(a')}/(1 + N_{s_t}(a))$, and $Q_{s_t}(a) = W_{s_t}(a) - L_{s_t}(a)$ with an exploration hyperparameter $c$. The virtual loss is also updated $L_{s_t}(A_t) \leftarrow L_{s_t}(A_t) + \nu$ using a virtual loss hyperparameter $\nu$. The virtual loss prevents the tree search from visiting the same state more than once and discourages the asynchronous workers from all following the same path (Segal, 2011).

Once a leaf node, $s_\tau$, is reached, the state is expanded by adding the children of $s_\tau$, $\{A(s_\tau, a), \forall a \in \mathcal{A}\}$, to the tree $T$. Then, for each child $s'$, the memories are initialized: $W_{s'}(\cdot) = 0$, $N_{s'}(\cdot) = 0$, $P_{s'}(\cdot) = \boldsymbol{p}_{s'}$, and $L_{s'}(\cdot) = 0$, where $\boldsymbol{p}_{s'}$ is the policy output of the network $f_\theta(s')$. Next, the value and policy are computed: $(v_{s_\tau}, \boldsymbol{p}_{s_\tau}) = f_\theta(s_\tau)$ and the value is backed up on all visited states in the simulated path. For $0 \le t \le \tau$, the memories are updated: $W_{s_t}(A_t) \leftarrow \max(W_{s_t}(A_t), v_{s_\tau})$, $N_{s_t}(A_t) \leftarrow N_{s_t}(A_t) + 1$, $L_{s_t}(A_t) \leftarrow L_{s_t}(A_t) - \nu$. Note that, unlike other implementations of MCTS, only the maximal value encountered along the tree is stored, and not the total value. This is because the Rubik's Cube is deterministic and not adversarial, so we do not need to average our reward when deciding a move.

The simulation is performed until either $s_\tau$ is the solved state or the simulation exceeds a fixed maximum computation time. If $s_\tau$ is the solved state, then the tree $T$ of the simulation is extracted and converted into an undirected graph with unit weights. We expand the graph by adding the children

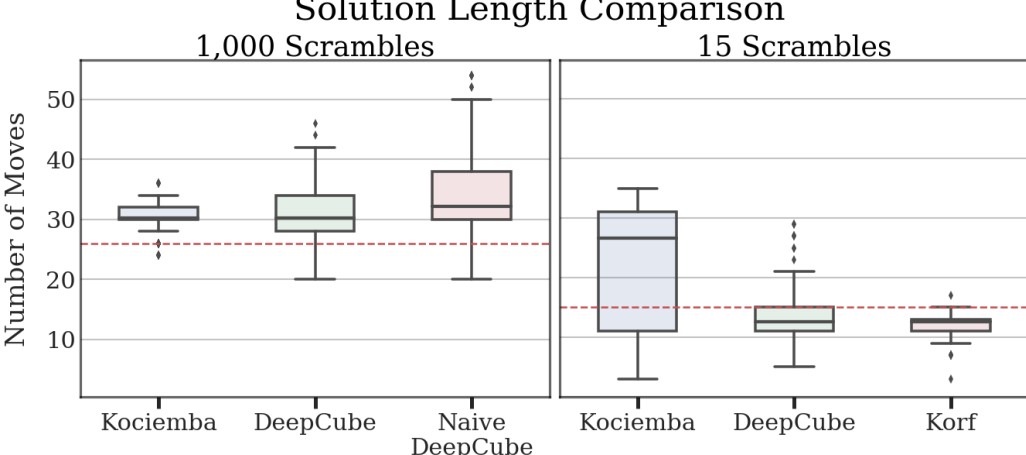

Figure 4: Distribution of solution lengths for DeepCube, Kociemba, and Korf. The left graph features naive DeepCube to evaluate the effect of our shortest path search. The right graph features the Korf optimal solver to evaluate how well DeepCube can find short solutions. The red lines represent the 26 and 15 move upper bound on the left and right respectively.

of any unexplored states and we perform a full breath-first search on $T$ to find the shortest predicted path from the starting state to solution. Alternatively, the last sequence $Path = \{A_t | 0 \leq t \leq \tau\}$ may be used, but this produces longer solutions.

## 5   RESULTS

As a baseline, we compare DeepCube against two other solvers. The first baseline is the **Kociemba** two-stage solver  (Kociemba; Tsoy, 2018). This algorithm relies on human domain knowledge of the group theory of the Rubik's Cube. Kociemba will always solve any cube given to it, and it runs very quickly. However, because of its general-purpose nature, it often finds a longer solution compared to the other solvers. The other baseline is the **Korf** Iterative Deepening A* (IDA*) with a pattern database heuristic  (Korf, 1997; Brown, 2017). Korf's algorithm will always find the optimal solution from any given starting state; but, since it is a heuristic tree search, it will often have to explore many different states and it will take a long time to compute a solution. We also compare the full DeepCube solver against two variants of itself. First, we do not calculate the shortest path of our search tree and instead extract the initial path from the MCTS: this will be named **Naive DeepCube**. We also use our trained value network as a heuristic in a greedy best-first search for a simple evaluation of the value network: this will be named **Greedy**.

We compare our results to Kociemba using 640 randomly scrambled cubes. Starting from the solved cube, each cube was randomly scrambled 1000 times. Both DeepCube and Kociemba solved all 640 cubes within one hour. Kociemba solved each cube in under a second, while DeepCube had a median solve time of 10 minutes. The systematic approach of Kociemba explains its low spread of solution lengths with an IQR of only 3. Although DeepCube has a much higher variance in solution length, it was able to match or beat Kociemba in 55% of cases. We also compare DeepCube against Naive DeepCube to determine the benefit of performing the BFS on our MCTS tree. We find that the BFS has a slight, but consistent, performance gain over the MCTS path ($-3$ median solution length). BFS removes any trivial cycles in the path such as a move followed by its inverse, but it also improves solution lengths by finding a slightly more efficient path between states that are within 3 moves of each other. A comparison of solution length distributions for these three solvers is presented in the left graph of Figure 4.

We could not include Korf in the previous comparison because its runtime is prohibitively slow: solving just one of the 640 cubes took over 6 days. We instead evaluate the optimality of solutions found by DeepCube by comparing it to Korf on cubes closer to solution. We generated a new set of 100 cubes that were only scrambled 15 times. At this distance, all solvers could reliably

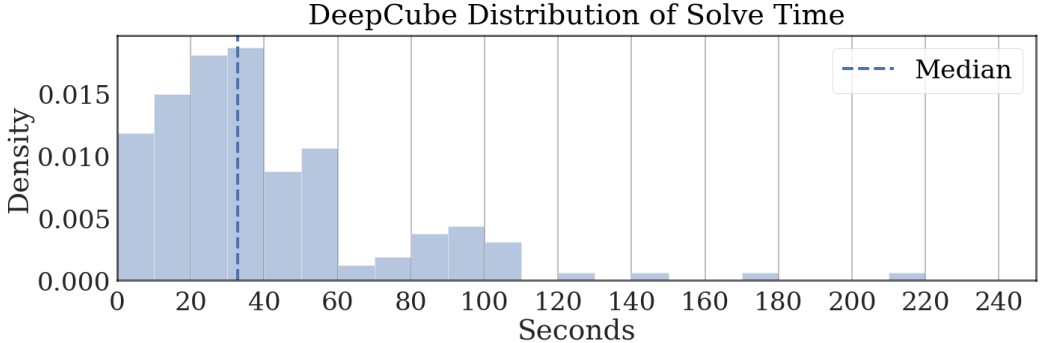

solve all 100 cubes within an hour. We compare the length of the solutions found by the different solvers in the right graph of Figure 4. Noticeably, DeepCube performs much more consistently for close cubes compared to Kociemba, and it almost matches the performance of Korf. The median solve length for both DeepCube and Korf is 13 moves, and DeepCube matches the optimal path found by Korf in 74% of cases. However, DeepCube seems to have trouble with a small selection of the cubes that results in several solutions being longer than 15 moves. Note that Korf has one outlier that is above 15 moves. This is because Korf is based on the half-turn metric while we are using the quarter-turn metric. Furthermore, our network also explores far fewer tree nodes when compared to heuristic-based searches. The Korf optimal solver requires an average expansion of 122 billion different nodes for fully scrambled cubes before finding a solution (Korf, 1997). Our MCTS algorithm expands an average of only 7,823 nodes with a maximum of 24,175 expanded nodes on the longest running sample. This is why DeepCube is able to solve fully scrambled cubes much faster than Korf. Furthermore, this small amount of nodes explored means that the algorithm can potentially be more efficient with an implementation that better exploits parallel processing.

## 6    DISCUSSION

DeepCube is based on similar principles as AlphaZero and ExIt, however, these methods receive their reward when the game reaches a terminal state, which is guaranteed to occur given enough play time. On the other hand, one is not guaranteed to find a terminal state in the Rubik's Cube environment and therefore may only encounter rewards of -1, which does not provide enough information to solve the problem. DeepCube addresses this by selecting a state distribution for the training set that allows the reward to be propagated from the terminal state to other states. In addition, while AlphaZero and ExIt make use advanced tree-search algorithms to update the policy, DeepCube is able to find success using the faster and simpler depth-1 BFS.

The depth-1 BFS used to improve the policy can also be viewed as doing on-policy temporal difference learning (Sutton, 1988), specifically TD(0) with function approximation. In this case, the TD(0) algorithm uses a deterministic greedy policy where each episode is one step long. Off-policy methods are often used to train a deterministic policy by using a stochastic behavior policy to facilitate exploration. An alternative approach, called exploring starts, instead changes the distribution of starting states and can also be used to train a deterministic policy in an on-policy fashion (Sutton & Barto, 1998). We use a similar approach to exploring starts by ensuring exploration through the selection of the starting state distribution.

The Rubik's Cube can be thought of as a classical planning problem. While traditional planning algorithms, such as Dijkstra's algorithm, would require an infeasible amount of memory to work on environments a state space as large as the Rubik's Cube, we show that Dual Policy Iteration can find a solution path in such an environment. For future work, we look to apply Autodidactic Iteration to a variety of other problems with similar characteristics such as robotic manipulation, two-player games, and path finding.

Léon Bottou defines reasoning as "algebraically manipulating previously acquired knowledge in order to answer a new question"(Bottou, 2013). Many machine learning algorithms do not reason about problems but instead use pattern recognition to perform tasks that are intuitive to humans, such as object recognition. By combining neural networks with symbolic AI, we are able to create

algorithms which are able to distill complex environments into knowledge and then reason about that knowledge to solve a problem. DeepCube is able to teach itself how to reason in order to solve a complex environment with only one positive reward state using pure reinforcement learning.

## 7 ACKNOWLEDGEMENTS

This work was supported in part by NSF Grant 1839429.

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

## A KNOWLEDGE LEARNED

DeepCube discovered a notable amount of Rubik's Cube knowledge during its training process, including the knowledge of how to use complex permutation groups and strategies similar to the best human "speed-cubers". For example, DeepCube heavily uses one particular pattern that commonly appears when examining normal subgroups of the cube: $aba^{-1}$. That is, the sequences of moves that perform some action $a$, performs a different action $b$, and then reverses the first action with $a^{-1}$. An intelligent agent should use these conjugations often because it is necessary for manipulating specific cubelets while not affecting the position of other cubelets.

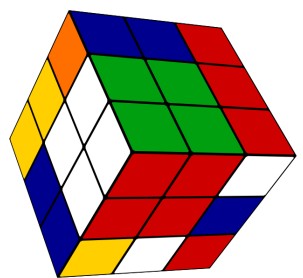

We examine all of the solutions paths that DeepCube generated for the 640 fully scrambled cubes by moving a sliding window across the solutions strings to gather all triplets. We then compute the frequency of each triplet and separate them into two categories: matching the conjugation pattern $aba^{-1}$ and not matching it. We find that the top 14 most used triplets were, in fact, the $aba^{-1}$ conjugation. We also compare the distribution of frequencies for the two types of triplets. In Figure 6, we plot the distribution of frequencies for each of the categories. We notice that conjugations appear consistently more often than the other types of triplets.

Figure 5: An example of Deep-Cube's strategy. On move 17 of 30, DeepCube has created the 2x2x2 corner while grouping adjacent edges and corners together.

We also examine the strategies that DeepCube learned. Often, the solver first prioritizes completing a 2x2x2 corner of the cube. This will occur approximately at the half way point in the solution. Then, it uses these conjugations to match adjacent edge and corner cubelets in the correct orientation, and it returns to either the same 2x2x2 corner or to an adjacent one. Once each pair of corner-edge pieces is complete, the solver then places them into their final positions and completes the cube. An example of this strategy is presented in Figure 5. This mirrors a strategy that advanced human "speed-cubers" employ when solving the cube, where they prioritize matching together corner and edge cubelets before placing them in their correct locations.

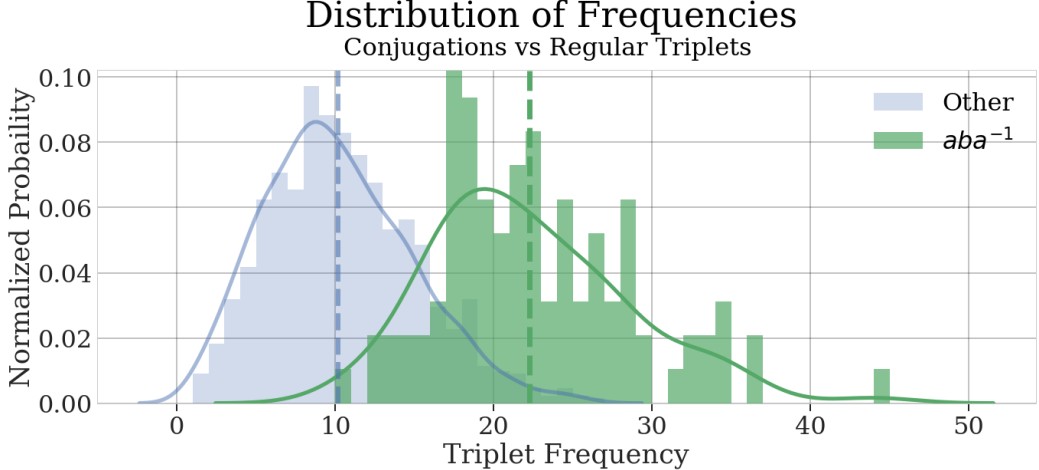

Figure 6: Comparison of the distribution of frequency of the two types of triplets. We split the triplets into conjugations ($aba^{-1}$) and non-conjugations. We then calculate the frequency of each triplet and plot the two distributions. The two vertical lines are the means of their respective distributions.

## B  TRAINING AND SOLVER DETAILS

We used a feed forward network as the architecture for $f_\theta$ as shown in Figure 7. The outputs of the network are a 1 dimensional scalar $v$, representing the value, and a 12 dimensional vector $\boldsymbol{p}$, representing the probability of selecting each of the possible moves. The network was then trained using ADI for 2,000,000 iterations. The network witnessed approximately 8 billion cubes, including repeats, and it trained for a period of 44 hours. Our training machine was a 32-core Intel Xeon E5-2620 server with three NVIDIA Titan XP GPUs.

During play, the neural network prediction is the major bottleneck for performance. In order to counteract this, we implemented a parallel version of MCTS that had 32 independent workers that shared a single MCTS search tree. Each of the workers queue their prediction requests, and the neural network batches all requests and processes them simultaneously. This parallelization sped up the solver 20x compared to a single core implementation.

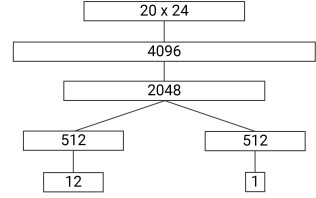

Figure 7: Architecture for $f_\theta$. Each layer is fully connected. We use *elu* activation on all layers except for the outputs. A combined value and policy network results in more efficient training compared to separate networks. (Silver et al., 2017b).

## C  OVERALL SOLVER COMPARISON

| Solver | Nodes | Seconds | Nodes/Sec | Mean Solution Length | Memory Requirement |
|--------|-------|---------|-----------|----------------------|--------------------|
| Rokicki | 1.86E+06 | **2.74** | **1.86E+06** | **20.6** | 182 GB |
| Korf | 122 billion | 40 | 2E+06 | **20.6** | 2 GB |
| DeepCube | **7.823E+03** | 40 | 2.1E+03 | 30.6 | 1 GB |
| Kociemba | N/A | 0.035 | N/A | 30.5 | 30 MB |

