# OpenReview forum: "Solving the Rubik's Cube with Approximate Policy Iteration"
_ICLR.cc/2019/Conference_

### Official Review · AnonReviewer2 · 2018-10-31
**interesting deep-RL tweaks to solve problem with sparse reward**

**Rating:** 7
**Confidence:** 3

**Review:**

This paper introduces a deep RL algorithm to solve the Rubik's cube. The particularity of this algorithm is to handle the huge state space and very sparse reward of the Rubik's cube. To do so, a) it ensures each training batch contains states close to the reward by scrambling the solution; b) it computes an approximate value and policy for that state using the current model and c) it weights data points based by the inverse of the number of random moves from the solution used to generate that training point. The resulting model is compared to two non-ML algorithms and shown to be competitive either on computational speed or on the quality of the solution.

This paper is well written and clear. To the best of my knowledge, this is the first RL-based approach to handle the Rubik's cube problem so well. The specificities of this problem make it interesting. While the idea of starting from the solution seemed straightforward at first, the paper describes more advanced tricks claimed to be necessary to make the algorithm work. The algorithm seems to be quite successful and competitive with expert algorithms, which I find very nice. Overall, I found the proposed approach interesting and sparsity of reward is an important problem so I would rather be in favor of accepting this paper.

On the negative side, I am slightly disappointed that the paper does not link to a repository with the code. Is this something the authors are considering in the future? While it does not seem difficult to code, it is still nice to have the experimental setup.

There has been (unsuccessful) attempts to solve the Rubik's cube using deep RL before. I found some of them here: https://github.com/jasonrute/puzzle_cube . I am not sure whether these can be considered prior art as I could not find associated accepted papers but some are quite detailed. Some could also provide additional baselines for the proposed methods and highlight the challenges of the Rubik's cube.

I am also curious whether/how redundant positions are handled by the proposed approach and wished this would be discussed a bit. Considering the nature of the state space and the dynamics, I would have expected this to be a significant problem, unlike in Go or chess. Does the algorithm forbid the reverse of the last action? Is the learned value/policy function good enough that backwards moves are seldom explored? Since the paper mention that BFS is interesting to remove cycles, I assume identical states are not duplicated. Is this correct?

---

> ### Author Response · Authors · 2018-11-27
> **Response to Reviewer 3**
>
> We would like to thank the reviewer for their helpful comments and for pointing us to the github resource.
>
> > “I am slightly disappointed that the paper does not link to a repository with the code. Is this something the authors are considering in the future?”
>
> We fully agree with the reviewer that releasing the code is important. We plan to release the code if the paper gets accepted. We have not done so yet to maintain anonymity.
>
> > “I am also curious whether/how redundant positions are handled by the proposed approach...Does the algorithm forbid the reverse of the last action? Is the learned value/policy function good enough that backwards moves are seldom explored? Since the paper mention that BFS is interesting to remove cycles, I assume identical states are not duplicated. Is this correct?”
>
> We did not strictly forbid reverse moves during the search. However, because we penalize longer solutions, because MCTS attempts many paths simultaneously, and because the virtual loss prevents duplicate exploration, the solver rarely explored repeat states. The BFS expansion of the path was a post-processing step we applied to the resulting path to obtain slightly better solutions. Although this did remove duplicates (if they existed), it more importantly allowed us to find "shortcuts" within our path. For example, we can replace say a 7-move sequence with a slightly more efficient 5-move sequence that MCTS didn't find. This effect was minimal but consistent.

---

### Official Review · AnonReviewer3 · 2018-11-03
**Nice idea but little study**

**Rating:** 7
**Confidence:** 4

**Review:**

The authors show how to solve the Rubik cube using reinforcement learning (RL) with Monte-Carlo tree search (MCTS). As common in recent applications like AlphaZero, the RL part learns a deep network for policy and a value function that reduce the breadth (policy) and depth (value function) of the tree searched in MCTS. This basic idea without extensions fails when trying to solve the Rubik cube because there is only one final success state so the early random policies and value functions never reach it. The solution proposed by the authors, called autodidactic iteration (ADI) is to start from the final state, construct a few previous states, and learn value function on this data where in a few moves a good state is reached. The distance to the final state is then increased and the value function learn more and more. This is an interesting idea that solves the Rubik cube, but the paper lacks a more detailed study. What other problems can be solved like this? Would a single successful trajectory be enough to use it in a wider context (as in https://blog.openai.com/learning-montezumas-revenge-from-a-single-demonstration/) ? Is the method to increase distance from final state specific to Rubik cube or general? Is the training stable with respect to this or is it critical to get it right? The lack of analysis and ablations makes the paper weaker.

[Revision] Thanks for the replies. I still believe experiments on more tasks would be great but will be happy to accept this paper.

---

> ### Author Response · Authors · 2018-11-27
> **Response to Reviewer 2**
>
> We would like to thank the reviewer for their helpful comments.
>
> > “What other problems can be solved like this?”
>
> This approach can be used in two different types of problems. The first is planning problems in environments with a high number of states. The second type of problem is when you need to find one specific goal but might not know what the goal is. However, if you have examples of solved examples you can train a value function using ADI on these solved examples and hopefully it will transfer to the new problems. For instance, in protein folding, the goal is to find the protein conformation with minimal free energy. We don’t know what the optimal conformation is beforehand, but we can train a value network using ADI on proteins where we know what their optimal conformation is.
>
>
> > “Would a single successful trajectory be enough to use it in a wider context? (as in https://blog.openai.com/learning-montezumas-revenge-from-a-single-demonstration/)”
>
> For our method to work, all we need is the ability to start from the goal state and take moves in reverse. Therefore, not only is a single successful trajectory sufficient, all that is needed is the final state of that successful trajectory: the goal state. Using only the goal state, it can generate other states by randomly taking actions away from the goal state.
>
> > “Is the method to increase distance from final state specific to Rubik cube or general?”
>
> The core concept is that the agent uses dynamic programming to propagate knowledge from easier examples to more difficult examples. Therefore, this method is applicable to any scenario in which one can generate a range of states whose difficulty ranges from easy to hard. For our method, we achieved this by randomly scrambling the cube 1 to N times. There has been other work in the field of robotics [1], as well as the work on Montezuma’s Revenge provided by the reviewer, that builds a curriculum starting by first generating states close to the goal and then progressively increasing the difficulty as performance increases. Instead of adaptively changing the state distribution during training, our method fixes the state distribution before training while the targets for the state values change as the agent learns.
>
> > “Is the training stable with respect to this or is it critical to get it right?”
>
> We found that the value of N, the maximum number of times to scramble the solved cube, was not crucial to the stability of training. It only had an effect on the final performance. If N was too low (e.g. 5), then DeepCube only performed well on cubes close to the solutions, but not on more complicated cubes. If N was too high (e.g. 100), then it took more iterations to learn; nonetheless, the agent would still learn. We found that N=30 resulted in both good value function estimation as well as reasonable training time.
>
> [1] Florensa, C., Held, D., Wulfmeier, M., Zhang, M., & Abbeel, P. (2017). Reverse curriculum generation for reinforcement learning. arXiv preprint arXiv:1707.05300.

---

### Official Review · AnonReviewer1 · 2018-11-03
**A good paper**

**Rating:** 7
**Confidence:** 4

**Review:**

The authors provide a good idea to solve Rubik’s Cube using an approximate policy iteration method, which they call it as Autodidactic iteration. The method overcomes the problem of sparse rewards by creating its own rewards system. Autodidactic iteration starts with solved cube and then propagate backwards to the state.

The testing results are very impressive. Their algorithm solves 100% of randomly scrambled(1000 times) cubes and has a median solve length of 30 moves. The God’s number is 26 in the quarter turn metric, while their median moves 30 is only 4 hands away from the God’s number. I appreciate the non-human domain knowledge part most because a more general algorithm can be used to other area without  enough pre-knowledges.

The training conception to design rewards by starting from solved state to expanded status is smart, but I am not very clear how to assign the rewards based on the stored states? Only pure reinforcement learning method applied sounds simple, but performance is great. The results are good enough with the neural network none-random search guidance. Do you have solving time comparison  between your method and other approximate methods?

Pros: -  solved nearly 100% problems with reasonable  moves.
          -  a more general algorithm solving unknown states value problems.

Cons: - the Rubik’s cube problem has been solved with other optimal approaches in the past. This method is not as competitive as other optimal solution solver within similar running time for this particular game.
           - to solve more dimension cubes, this method might be out of time.

---

> ### Author Response · Authors · 2018-11-27
> **Response to Reviewer 1**
>
> We would like to thank the reviewer for their helpful comments.
>
> > “I am not very clear how to assign the rewards based on the stored states?”
>
> The environment returns a reward of +1 for the solved state and a reward of -1 for all other states. From this single positive reward given at the solved state, DeepCube learns a value function. Using dynamic programming, DeepCube improves its value estimate by first learning the value of states one move away from the solution and then building off of this knowledge to improve its value estimate for states that get progressively further away from the solution.
>
> > “Do you have solving time comparison between your method and other approximate methods?”
>
> Yes, we have improved the efficiency of our solver since we last submitted our paper by optimizing our code. Our method takes, on average, 40 seconds; whereas the fastest optimal solver we could find (implemented by Tomas Rokicki to find “God’s number” [1]) for the Rubik’s Cube takes 2.7 seconds. These results are summarized in section C of the appendix of the updated paper. While Rokicki’s algorithm is faster, Rokicki’s algorithm also uses knowledge of groups, subgroups, cosets, symmetry, and pattern databases. On the other hand, our algorithm does not exploit any of this knowledge and learns how to solve the Rubik’s Cube given only basic information about the problem. In addition, Rokicki’s solver uses 182GB of memory to run whereas ours uses at most 1GB. These differences are summarized in the updated paper. We are currently making better use of parallel processing and memory to improve the speed of our algorithm.
>
> [1] Rokicki, T., Kociemba, H., Davidson, M., & Dethridge, J. (2014). The diameter of the Rubik's Cube group is twenty. SIAM Review, 56(4), 645-670.

---

> > ### Public Comment · (anonymous) · 2019-01-22
> > **Not Really Sparse Rewards?**
> >
> > If it is receiving a reward of -1 for all other states, isn't this no longer a sparse reward problem? It is getting feedback for every move taken, which biases it against getting into cycles, and towards shorter solution lengths (in episodic problems).
> >
> > Also, how long was Kociemba's solver given to run? Doesn't that solver eventually converge on the optimal solution, or was the first outputted solution what was compared to? This solver is widely used by the speedcubing community and it generally finds solutions in the low 20s or better within a couple of seconds. The algorithm itself states that the worse-case scenario of Kociemba is 30 moves, yet the graph seems to show solution lengths longer than this.
> >
> > There is also relatively recent work in bi-directional search which has been capable of optimally solving the hardest positions (as opposed to just proving the solution length is <= 20 moves) reasonably quickly with far fewer node expansions- this might be useful to compare to.

---

### Meta-Review · Area_Chair1 · 2018-12-14
**Interesting work, but too focused on a particular problem**

**Confidence:** 4
**Recommendation:** Accept (Poster)

**Metareview:**

The paper introduces a version of approximate policy iteration (API), called Autodidactic Iteration (ADI), designed to overcome the problem of sparse rewards.  In particular, the policy evaluation step of ADI is trained on a distribution of states that allows the reward to easily propagate from the goal state to states farther away.  ADI is applied to successfully solve the Rubik's Cube (together with other existing techniques).

This work is an interesting contribution where the ADI idea may be useful in other scenarios.  A limitation is that the whole empirical study is on the Rubik's Cube; a controlled experiment on other problems (even if simpler) can be useful to understand the pros & cons of ADI compared to others.

Minor: please update the bib entry of Bottou (2011).  It's now published in MLJ 2014.